# Mast Cells and Interleukins

**DOI:** 10.3390/ijms232214004

**Published:** 2022-11-13

**Authors:** Antonio Giovanni Solimando, Vanessa Desantis, Domenico Ribatti

**Affiliations:** 1Guido Baccelli Unit of Internal Medicine, Department of Biomedical Sciences and Human Oncology, School of Medicine, Aldo Moro University of Bari, 70121 Bari, Italy; 2Medical Oncology Unit, IRCCS Istituto Tumori “Giovanni Paolo II” of Bari, 70124 Bari, Italy; 3Department of Biomedical Sciences and Human Oncology, Pharmacology Section, University of Bari Medical School, 70121 Bari, Italy; 4Department of Translational Biomedicine Medicine and Neuroscience, University of Bari Medical School, 70124 Bari, Italy

**Keywords:** inflammatory cells, interleukins, mast cells, tumors

## Abstract

Mast cells play a critical role in inflammatory diseases and tumor growth. The versatility of mast cells is reflected in their ability to secrete a wide range of biologically active cytokines, including interleukins, chemokines, lipid mediators, proteases, and biogenic amines. The aim of this review article is to analyze the complex involvement of mast cells in the secretion of interleukins and the role of interleukins in the regulation of biological activities of mast cells.

## 1. Introduction

Mast cells are present in all classes of vertebrates and emerged >500 million years ago, long before the development of adaptive immunity [1]. Mast cells are bone marrow-derived cells which were identified by Paul Ehrlich and first described in his doctoral thesis in 1878 [2]. Each cell has a single round or oval-shaped nucleus, and the cytoplasm contains numerous secretory granules that metachromatically stain with thiazine dyes such as toluidine blue. The metachromatic staining of mast cell granules probably reflects their content of proteoglycans, such as chondroitin sulphates and heparin.

Mast cells originate in the bone marrow microenvironment following the myeloid pathway (Figure 1). The first evidence concerning their origin from bone marrow precursors of the hematopoietic lineage was reported by Kitamura and co-workers in 1977 by means of experiments involving the adaptive transfer of wild-type bone marrow into mast cell-deficient mice. Recent works indicate that mast cells can be derived from erythro-myeloid progenitors (EMPs) which originate in the yolk sac before the emergence of hematopoietic stem cells. In this context, perturbation of the maternal environment leads to the production of mediators such as cytokines that cause epigenetic alterations in EMPs and thereby the chronic activation of EMP-derived immune cells, including mast cells, after birth [3].

Mast cells move to different organs through the vascular system, where they differentiate under the control of factors released by the local specific microenvironments [4]. In this context, mast cells are present in most tissues and organs of the human body. The principal function of these cells is to impede the body’s interaction with different pathogens. As such, they are generally strategically located at the interface between inner organs and outer microenvironments in the skin, respiratory, gastro-intestinal, and urogenital tracts. Moreover, mast cells are localized in connective tissues; are strictly related to blood, lymphatic vessels, and nerves; and are involved in different processes, including wound healing, tissue regeneration and remodeling, fibrosis, autoimmunity, tumor growth, and angiogenesis.

Mast cells are involved in different processes beyond their classical role in Ig-E-mediated allergic reactions [5]. They participate, together with other effector cells, in many different innate and adaptive immunological responses against pathogens. Mast cells release different anti- and pro-inflammatory agents, changing their role from protective to pro-inflammatory cells involved in different pathological conditions.

Mast cells can exert multiple actions in many pathological conditions through a plethora of mediators contained in their cytoplasmic granules which, after mast cell stimulation, are released into the extracellular space through different mechanisms of degranulation (Figure 2).

Mast cell mediators are divided into three general groups: (i) pre-formed mediators, (ii) de novo synthesized lipid mediators, and (iii) cytokines/chemokines. Pre-formed mediators, include histamines, proteases, serotonin, heparin, proteases proteoglycans, and tumor necrosis factor (TNF), among others. Human mast cells synthesize and release an array of cytokines and growth factors involved in inflammation, immunity, hematopoiesis, tissue remodeling, and other biological functions [6]. Among these neo-synthesized mediators, there are different classes of interleukins (ILs), which are primarily secreted by leukocytes and macrophages and regulate numerous biological processes and immune response.

ILs are involved in different processes, including the activation and differentiation of immune cells and their proliferation, maturation, migration, and adhesion during inflammatory processes and immune responses. They modulate the biological activities of different cells through the stimulation or the inhibition of the secretion of other cytokines. ILs include over 60 different molecular families of cytokines, which can be divided into four main groups: the IL-1-like cytokines, the class 1 helical cytokines (IL-4-like, γ-chain, and IL-6/IL-12- like), the class 2 helical cytokines (IL-10-like and IL-28-like), and the IL-17-like cytokines [7].

ILs are key regulators of the tumor microenvironment and are responsible for tumor–immune cell crosstalk. In this context, ILs are relevant in the development and progression of cancer. Although various immune effector cells, including mast cells, are recruited to the tumor site, their antitumor functions are downregulated, largely in response to tumor-derived factors or signals [8].

A multitude of cellular sources, including mast cells, receptors, and signaling pathways, are responsible for the pleiotropic role of ILs in cancer. Different components of the mast cell secretome, including secreted de novo synthesized soluble molecules, secreted preformed granules, and membrane-enclosed extracellular vesicles, are involved in the release of ILs into the tumor microenvironment [9].

The aim of this article is to analyze the involvement of mast cells in the secretion of ILs and the role of ILs in the regulation of the biological activities of mast cells in autoimmune diseases and tumor growth.

## 2. Interleukins 1 to 8

IL-1 is involved in the immune response mediated by dendritic cells, monocytes, and macrophages. IL-1 exerts a pro-inflammatory action, inducing an increase in vascular permeability at the site of secretion, with a consequent increase in the concentration of leukocyte migration at the anatomical site of inflammation. Mast cells synthesize and secrete high levels of IL-1, which is also involved in lymphatic infiltration [9]. Mast cell-derived IL-1β is involved in the development of different models of skin inflammation [10] and arthritis [11] in mice [12]. Mast cells stimulate the secretion of IL-1β by macrophages in the pathogenesis of rheumatoid arthritis [11]. Mast cell IL-2 contributes to the expansion of Treg cells involved in immune suppression in an experimental model of IL-33-induced airway inflammation [13].

IL-2, produced by T cells during an immune response, is necessary for the growth and differentiation of naïve T cells into effector T cells. It stimulates proliferation and enhances the biological activities of other T cells, NK cells, and B cells. The bone marrow immune microenvironment represents a crucial source of mast cells in different conditions [14,15,16]. IL-2 released by cultured mouse mast cells contributes to the suppression of chronic allergic dermatitis upon activation with IgE and antigen in vitro [17]. Mast cell IL-2 was also involved in a positive loop contributing to lung pathology in an experimental model of cystic fibrosis [18].

IL-3 is involved in the growth of immature progenitor cells of all hematopoietic lineages. Moreover, IL-3 promotes the survival of macrophages, mast cells, and megakaryocytes, and it is synthesized by lymphoid cells, mast cells, and eosinophils. Mast cells produce IL-3 upon IgE-mediated stimulation [19]. Mast cell-derived IL-3 promotes the differentiation and survival of other myeloid cells, including basophils, eosinophils, and macrophages [20].

IL-4 has the capacity to co-stimulate B cell growth in vitro and was initially named B cell stimulatory factor. IL-4 inhibits the macrophage production of TNF-α, IL-1, and IL-6. IL-4 also enhances the production of molecules that counteract these cytokines, such as IL-1R antagonist and the decoy receptor IL-1RII. Mast cell-released IL-4 triggers Th2 lymphocytes to synthesize IL-4 to induce inflammatory cell accumulation and B lymphocyte Ig switching to IgE [21]. Mast cells produce IL-4 upon IgE-mediated stimulation or in response to calcium ionophores [22], IL-33 [23], or certain lectins [24]. Mast cell-derived IL-4 activates CD4^+^ T cells [25] and induces IgE production by B cell-derived plasma cells [25]. Mast cells promote fibroblast proliferation through the release of IL-4 [26]. Moreover, mast cells release IL-4, which polarizes T cells to a Th2 phenotype, confirming the importance of T cells and mast cells interactions [27]. In this context, mast cell IL-4 stimulates T cell adhesion to the endothelial cells of fibroblasts, a process which is mediated by the expression of intercellular adhesion molecule-1 (ICAM-1), endothelial-leukocyte adhesion molecule-1 (ELAM-1), and vascular cell adhesion molecule-1 (VCAM-1) [28]. IL-4 immunoreactive mast cells have been detected in bioptic specimens of allergic rhinitis [21], asthma [29], and atopic dermatitis [30]. It is unclear whether mast cell IL-4 released extracellularly is due to degranulation or constitutive exocytosis [31].

IL-5 is involved in the eosinophil arm of the Th2 response, promoting their survival, differentiation, and chemotaxis. IL-5 activates mature eosinophils such that they can kill helminth worms via degranulation. Mast cells release IL-5 and IL-6 in human allergic inflammation [21]. Mouse and human mast cells produce IL-5 upon IgE-mediated stimulation [32]. IL-5 immunoreactive mast cells are detected in human duodenal [33], bronchial, and nasal biopsies [34], as well as in bioptic specimens of patients with asthma [33]. Mast cell-released IL-5 recruits eosinophils [35].

IL-6 is a T cell-derived lymphokine involved in the maturation of B cells in plasma cells. IL-6 induces the proliferation of murine pluripotent stem cells in synergy with IL-3 and stimulates megakaryopoiesis with stem cell factor and thrombopoietin. IL-6 in human mast cells cultured from CD34^+^ cord blood progenitors stimulates an increase in cell size, chymase-positivity, and intracellular histamine levels [36]. Mast cell-derived IL-6 stimulates monocyte-macrophage activation and neutrophils recruitment [37]. Mast cells release IL-6 following activation by both degranulating and non-degranulating stimuli [38]. IL-6 production by rat peritoneal mast cells is induced by bacterial lipopolysaccharide [39], substance P [40], and IL-1 [41]. Mast cell-derived IL-6 is important in the Th17 immune response [42]. Mast cells counteract regulatory T cell suppression through IL-6 and OX40/OX40L ligand contact-dependent interactions towards Th-17 cell differentiation [43]. Under IL-6 availability, mast cells revert Treg cell suppressive activity, triggering OX40 on their membrane, which changes Treg cells from immunosuppressive into pro-inflammatory IL-17-producing cells [43]. Finally, IL-6 is critical for mast cell survival; in fact, IL-6 neutralization decreases mast cell viability under hypoxic conditions [44].

IL-8 attracts neutrophils, basophils, and T cells. It is involved in neutrophil activation and is released from different inflammatory cells, including monocytes, macrophages, and neutrophils. IL-8 is also involved in several processes, including mitogenesis, the stimulation of angiogenesis, inflammation, chemotaxis, neutrophil degranulation, and calcium homeostasis. Mast cells generate IL-8 [45]. Human intestinal mast cells synthesize IL-6 and IL-8 at low levels without stimulation of the cells [46]. When stimulated with IL-1β, mast cells produce angiogenic IL-8 through a leukotriene B4 receptor-2 [47].

## 3. Interleukins 9 to 33

IL-9, which is primarily synthesized by T cells, acts as a T cell and mast cell growth factor. The production of IL-9 is associated with the Th2 phenotype in allergies, and IL-9 is involved in allergic reactions in asthmatic patients through an increased expression of T cells obtained from bronchoalveolar lavage fluid. Mucosal mast cells synthesize IL-9, which may contribute to an IgE-mediated food allergy in a mouse model [48]. Mast cells express the receptor for IL-9, and IL-9 enhances the growth of human mast cell progenitors [49]. IL-9 production by activated mast cells is enhanced by lipopolysaccharide [50].

IL-10 is an anti-inflammatory cytokine involved in immune responses elicited in autoimmunity and allergy though the inhibition of many effector cells; its concentration is inversely correlated with disease incidence and severity. IL-10 is synthesized by B cells, monocytes, dendritic cells, NK cells, and T cells, as well as by keratinocytes and hepatic stellate cells. It inhibits the secretion of IL-1, interferon gamma (IFN-γ), and TNF-α, and it is also involved in the expression of surface molecules responsible for antigen presentation. IL-10 is also involved in B cell differentiation in plasma cells and in antibody secretion. Overall, IL-10 plays a crucial role in the control and modulation of inflammatory processes.

Mast cells release IL-10, which exerts its effects on multiple immune cells [51,52]. Mast cell-derived IL-10 limits skin pathology in contact dermatitis and chronic irradiation with ultraviolet-B [53]. The secretions of IL-10 and IL-12 by mast cells are regulated by phosphatidylinositol 3-kinase [54]. The addition of IL-10 to bone marrow mast cells inhibits Fc-epsilon RI expression in mouse mast cells [55] and CD117 expression [56] and induces the expression of MMCP-1 and -2 [57]. Mast cell IL-10 reduces dendritic cell differentiation and enhances their capacity to reduce T cell proliferation and cytokine production [58]. IL-10 secreted by Treg cells inhibits Fc epsilon expression by mast cells [17]. IL-10 inhibits mast cell degranulation by suppressing mast cell IgE receptor expression [59].

IL-11 is characterized as a hematopoietic cytokine with thrombopoietic activity. It is expressed in the central nervous system (CNS) and the gastrointestinal tract. The hematopoietic actions of IL-11 include the stimulation and proliferation of primitive stem cells. IL-11 also acts in synergy with other cytokines to support the proliferation and differentiation of hematopoietic stem cells. Mast cells produce IL-11 in response to an IgE-mediated stimulus [60].

IL-13 is involved in the control of the different cells, including monocytes/macrophages, B cells, and endothelial and epithelial cells. IL-13-specific activity in allergic reactions concerns the modulation of mucus hypersecretion, airway hyperreactivity (AHR), and metaplasia, which are critical events in asthma that lead to the replacement of epithelial cells with goblet cells.

Mast cells synthesize IL-13 upon stimulation with IgE and antigens [61]. IL-13 production by activated mast cells is strongly enhanced by lipopolysaccharide [50], and mast cells also increase the synthesis of IgE by B cells through the release of IL-13 [58]. Microvesicle shedding is one of the secretory pathways involved in the release of IL-15 in mast cells [62].

IL-16 is a T cell chemoattractant (lymphocyte chemoattractant factor). IL-16 is synthesized by immune and parenchymal cells, including CD4 and CD8 T cells, eosinophils, macrophage/dendritic cells, mast cells, bronchial epithelium, and fibroblasts. In Crohn’s disease, mast cell-derived IL-16 can recruit lymphocytes from the blood stream [63], and mast cells are also a cellular source of IL-16 in asthma [64].

IL-17 is a pro-inflammatory cytokine involved in host defense as well as inflammatory and autoimmune disease. IL-17 is an important cytokine of T-helper 17 (TH17) cell subsets and has key roles in extracellular pathogen protection and stimulates the inflammatory response in autoimmune disorders. A strict interaction is observed between mast cells that produce TNF, which induces the activation of T regulatory and Th17 cells in the cervical lymph nodes [43]. Mast cells secrete IL-17A in different inflammatory diseases [65], including asthma [66] and chronic spontaneous urticaria [67]. IL-17 derived from myeloid-derived suppressor cells recruits IL-9-producing Treg cells, which in turn secrete the IL-9 involved in the survival of tumor resident mast cells [68]. Mast cells also secrete IL-17A in rheumatoid synovitis [69].

IL-24 is a member of the IL-20 family and is released by mast cells. It is involved in allergic airway inflammation, chronic allergic urticaria, allergic contact, and atopic dermatitis [70].

IL-31 is a cytokine that is produced not only in T cells but also in mast cells. It is thought to play a key role in inflammatory diseases and in the pathogenesis of itch in atopic dermatitis. IL-31 induces CCL1, CCL17, and CCL22 chemokines in atopic dermatitis-irritated skin [71].

IL-33 is expressed by endothelial and epithelial cells and is stored in high amounts within epithelial and endothelial barriers. Moreover, it is expressed by many other cell types following pro-inflammatory stimulation. IL-33 amplifies the polarization of macrophages to the M2 alternatively activated phenotype, which, in the context of the experimental autoimmune encephalomyelitis (EAE) model, is protective. In mast cells, IL-33 is a potent inducer of proinflammatory cytokines and chemokines, including IL-5, IL-13, CXCL8, and CCL2, and it is produced by mast cells and regulates IgE-dependent inflammation [72].

IL-1, released by activated keratinocytes and mast cells, stimulates skin macrophages to release IL-36, a powerful pro-inflammatory IL-1 family member. IL-36 mediates both innate and adaptive immunity, including in chronic pro-inflammatory diseases such as psoriasis [73].

## 4. Mast Cells, Interleukins, and Disease Pathobiology

### 4.1. Role of Mast Cells and Interleukins in Autoimmune Diseases

Increased mast cell density is associated with autoimmune diseases [74], including rheumatoid arthritis [75], type 1 diabetes [76], multiple sclerosis [77], EAE [78], systemic lupus erythematous [79], psoriasis [80], Guillain–Barré syndrome [81], bullous pemphigoid [82], Sjogren syndrome [83], chronic idiopathic urticaria [84], and experimental vasculitis [85].

Rheumatoid arthritis is a pathological condition in which chronic inflammation taking place within the synovial membrane leads to synovial hyperplasia, which is characterized by the infiltration of inflammatory cells, including lymphocytes and mast cells. The final event is a progressive, erosive destruction of cartilage and of the underlying bone. In rheumatoid arthritis, an overexpression of pro-inflammatory cytokines within the synovial tissue has been observed, and among these, TNF-α is the major player in maintaining the inflammatory process.

Mast cells are also present in normal synovium and are increased in the synovial tissue of rheumatoid arthritis patients. Synovial fibroblasts produce IL-33 [86], which activates mast cells to release IL-17 and IL-13 [87]. In this context, synovial mast cells have a bidirectional interaction with fibroblasts, whereby fibroblasts activate mast cells through IL-33, and activated mast cells in turn can activate synovial fibroblasts. This crosstalk can induce fibroblasts to invade underlying cartilage tissue. Finally, IL-33 exacerbates antigen-induced arthritis by activating mast cells [88].

EAE and multiple sclerosis are CD4^+^ T cell-mediated autoimmune diseases affecting the CNS. Mast cells play a role in the pathogenesis of both diseases. Mast cells produce cytokines that have been implicated in EAE disease, such as TNF-α and IL-4. In this context, T cells activate mast cells present within the leptomeninges and the dura mater to produce and release TNF-α, which in turn stimulates other inflammatory cells to release IL-4 and IL-9. Indeed, the neutralization of IL-9 and the deficiency of its receptor modulate EAE, also in correlation with a reduction in Th17 cells, the production of IL-6 by macrophages, and a decreased number of mast cells [77].

IL-17 is the major cytokine involved in the pathogenesis of psoriasis. In this context, targeting the IL-17 cytokine family (IL-17A, IL-17F, and IL-17 receptor A) may be considered as a potential therapeutic strategy in the treatment of psoriasis and psoriatic arthritis [89]. Most of the IL-17-producing cells in psoriatic skin are mast cells [90].

### 4.2. Role of Mast Cell Interleukins in Tumor Progression

Many tumors, including malignant melanoma, breast carcinoma, gastric cancer, and colorectal adenocarcinoma, are characterized by a strong host response involving different inflammatory cells, including macrophages, lymphocytes, and mast cells.

Chronic inflammation is a crucial event involved in tumor growth and progression. Different stromal and inflammatory cells act in concert with endothelial cells to create and maintain a particular microenvironment that allows for the development and diffusion of tumors. The synergistic activities of these cells are specific to the anatomical localization of the tumor sites, where they contribute to an efficient vascular supply for tumor growth in situ and an easy pathway to escape metastatic processes.

Mast cells can be recruited into the tumor microenvironment by different chemotactic molecules released by tumor cells. Among these chemoattractant factors, there is stem cell factor (SCF), which is also the main survival factor for mast cells. SCF is a powerful mast cell chemoattractant and activator that is responsible for the accumulation of mast cells at the periphery of tumors, at their interface with healthy tissues. Mast cell recruitment occurs via the migration of resident mast cells from neighboring healthy tissue or through the de novo recruitment of mast cell progenitors from the vasculature sites close to the tumor.

Mast cells can exert both anti-tumorigenic (Table 1) and/or pro-tumorigenic roles [91,92]. Accordingly, increased mast cell numbers have been correlated with either a good or poor prognosis in several human tumors. Recently, Atiakshin and co-workers carried out a detailed assessment of the mechanisms of the biogenesis and excretion of mast cell proteases in melanoma. They demonstrated that the development of melanoma was accompanied by the appearance of several pools in the tumor-associated mast cell population, with a predominant content of one or two specific proteases and with a low content or complete absence of others. In this context, mast cells expressing specific proteases in the tumor stroma apparently belong to a protective mechanism against tumor evasion from immunocompetent cells and can serve as favorable prognostic markers [93].

Mast cells can stimulate tumor cell expansion by releasing cytokines and growth factors, including fibroblast growth factor-2 (FGF-2), nerve growth factor (NGF), platelet-derived growth factor (PDGF), IL-10, and IL-8, into the tumor stroma. In addition, mast cells synthesize and store angiogenic factors, as well as matrix metalloproteinases (MMPs), which promote tumor vascularization and tumor invasiveness, respectively. Mast cells synthesize and release several molecules which exert either stimulatory or inhibitory effect on angiogenesis. However, pro-angiogenic factors exceed the record of anti-angiogenic factors [94]. In this context, mast cells may be considered as cells playing a crucial role in the amplification of the angiogenic response.

Moreover, mast cells are a major source of histamine, which can induce tumor cell proliferation through H1 receptors while suppressing the immune system through H2 receptors. Mast cells exert immunosuppressive activities through the release of different mediators, including IL-10, histamine, and TNF-α. Moreover, mast cells inhibit tumor cell growth, apoptosis, diffusion, and inflammation by releasing cytokines, including IL-1, IL-4, IL-6, TNF-α, and chondroitin sulphate. Finally, tryptase is responsible for both tumor cell disruption and inflammation by means of the activation of protease-activated receptors (PAR-1 and -2). Tryptase is involved in the pathogenesis of different inflammatory diseases of the cardiovascular, respiratory, digestive, reproductive, and nervous systems, and its participation in the development of these pathological conditions is the consequence of the activation of PAR-2 receptors by proteases [95]. Mast cells may be polarized through either pro-inflammatory or anti-inflammatory profiles. A pro-inflammatory profile is characterized by a low expression of IL-10, while an anti-inflammatory profile is characterized by a high expression of IL-10 [96].

Overall, mast cells promote tumor development by altering stroma–epithelial interactions, by inducing tumor angiogenesis and lymphangiogenesis, and by releasing different cytokines and growth factors, as evaluated by validated assays [94]. In solid tumors, mast cells may be localized in intra-tumoral or peri-tumoral areas, expressing of favorable or poor prognosis, respectively [94].

Mast cells, through the release of IL-1, IL-4, and IL-6, are involved in the elimination of tumor cells and in the rejection of tumors [104]. IL-1 is linked to tumorigenesis and tumor progression in terms of local growth, angiogenesis, macrophage recruitment, and metastatic processes [105].

On the contrary, mast cells, through the release of IL-8 and IL-10, promote the expansion of tumor mass [106]. Tumor-infiltrating mast cells, after IL-2 pre-operative induction therapy, are responsible for an improved clinical outcome in patients with malignant mesothelioma [107]. Mast cells in basal cell carcinoma express IL-8 [108]. The supernatants of mast cells after stimulation with calcium ionophore induce the production of IL-6 and IL-17 from tumor cells and increase the proliferation of primary cutaneous T cell lymphoma [109]. IL-9 is responsible of the survival of mast cells in the tumor microenvironment [110]. The up-regulation of IL-17-triggered tumor-infiltrating mast cells might be involved in the remodeling of the tumor microenvironment [111]. SCF released by tumor cells activates the cKIT receptor expressed on mast cells, increasing IL-17 expression and favoring inflammation and immune-suppression in the tumor microenvironment [112]. Mast cells promote tumor progression through the release anti-inflammatory cytokines, such as IL-10 [113]. IL-13-positive mast cells are present in the nodular sclerosis subtype of classical Hodgkin’s lymphoma and are associated with a higher degree of fibrosis [114]. Mast cells and IL-33 are also involved in the regulation of the immune response in gastrointestinal cancers [115].

## 5. Therapeutic Approaches

Therapeutic approaches used in diseases in which mast cell numbers are increased include tyrosine kinase inhibitors (midostaurin, nilotinib, imatinib, and dasatinib) to target the cKIT receptor action and mast cell tryptase inhibitors (gabexate mesylate, nafamostat mesylate, and tranilast) [116]. At least in humans, mast cells seem to depend on the activation of KIT for survival. The first tyrosine kinase inhibitor introduced into the clinic, STI571 (imatinib mesylate, Gleevec), has inhibitory activity against the signaling cascade activated by the KIT receptor (CD117) [117]. This inhibitory activity is the basis of the use of imatinib against gastrointestinal stromal tumors (GIST) and against metastatic melanoma with cKIT mutations [118,119].

Mast cell-stabilizing drugs inhibit the release of mediators from mast cells and are used to prevent allergic reactions to common allergens. Mast cell-stabilizing agents, such as cromolyn sodium, have been used in different pre-clinical models of solid tumors. In a xenograft mouse model of thyroid cancer, treatment with cromolyn reduced tumor growth [120].

ILs may be considered as potential therapeutic targets in different pathological conditions, including autoimmune diseases and tumors. This is a consequence of the role that ILs have in the modulation of inflammatory and immune responses occurring in such pathological conditions. In this context, different agents against ILs are approved for mast cell-targeted biological treatments. Dupilumab, a monoclonal antibody (mAb) and a dual inhibitor of IL-4 and IL-13, has been approved for the treatment of atopic dermatitis [121] and asthma [122]. The anti-IL-5 mAbs Reslizumab [123] and Mepolizumab [124] and the anti-IL-5 receptor alpha mAb Benzalizumab [125] are used for the treatment of severe eosinophilic asthma. Nemolizumab, an mAb to IL-31 receptor alpha, led to significant improvements in pruritus and skin lesions in patients with atopic dermatitis [126]. IL-37 is an anti-inflammatory cytokine which binds to the α -chain of the IL-18 receptor α (IL-18Rα). IL-37 down-regulates TLR-MyD88 signaling in mast cells and may be used to inhibit its activation by IL-1 and IL-33 in Sįögren syndrome [127] and in skin inflammation in psoriasis [73]. Moreover, IL-37 exerts a protective role in allergic contact dermatitis through mast cell inhibition [128].

## Figures and Tables

**Figure 1 ijms-23-14004-f001:**
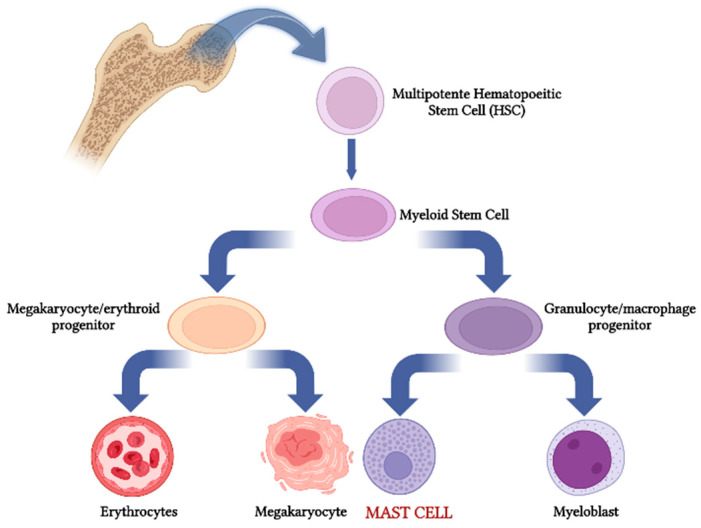
Mast cell differentiation along the myeloid pathway.

**Figure 2 ijms-23-14004-f002:**
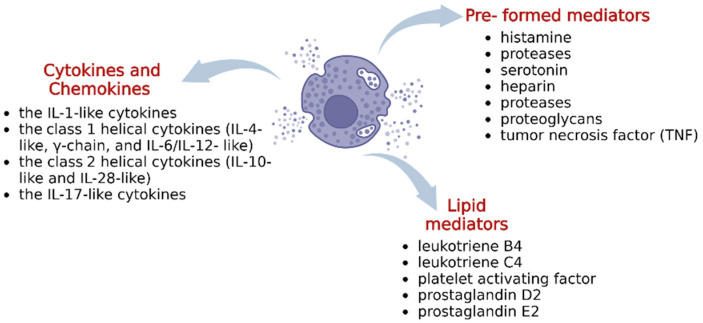
Mast cell secretome.

**Table 1 ijms-23-14004-t001:** Tumors in which mast cells exert anti-tumorigenic roles.

Type of Tumor	Mechanism	Reference
Ovarian cancer	Cytotoxicity, tumor growth inhibition	[97]
B cell lymphoma	Cytotoxicity, tumor growth inhibition	[98]
Melanoma	Cytotoxicity, tumor growth inhibition	[99]
Non-small cell lung cancer	Cytotoxicity, tumor growth inhibition	[100]
Prostate cancer	Tumor growth inhibition, tumor cell apoptosis	[101]
Colorectal cancer	Cytotoxicity, tumor growth inhibition	[102]
Breast cancer	Cytotoxicity, tumor growth inhibition	[103]

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
