# Peer review of "Mast Cells and Interleukins"

_ijms, 2022, doi:10.3390/ijms232214004_

Round 1

Reviewer 1 Report

This is a fantastic review describing the role of mast cells and its secretion of interleukins. Authors nicely review key recent articles to discuss findings about mast cells and interleukins in pathobiology of autoimmune diseases, and tumor progression. Additionally, this manuscript also describe therapeutic approaches involving mast cells and its secretome- interleukin. Following see my suggestions,

1.     Readers of this review article would be highly benefited if authors could include a graphical representation of secretome of mast cells and its involvement in diseases. This could include- a) pre-formed mediators, b) de-novo synthesized lipid mediators, and c) cytokines/chemokines with a major focus towards interleukins.

2.     A graphical representation of mast cell origin and differentiation could also be included here.

3.     Please add publications section to table 1.

4.     Authors should check font type, font style, line spacing thorough out the manuscript as it varies from one section to another.

Author Response

Readers of this review article would be highly benefited if authors could include a graphical representation of secretome of mast cells and its involvement in diseases. This could include- a) pre-formed mediators, b) de-novo synthesized lipid mediators, and c) cytokines/chemokines with a major focus towards interleukins.

We have added a new Figure 2, showing secretome of mast cells.

A graphical representation of mast cell origin and differentiation could also be included here.

We have added a new Figure 1, showing mast cell origin and differentiation.

Please add publications section to table 1.

We have added publications

Authors should check font type, font style, line spacing thorough out the manuscript as it varies from one section to another.

Done.

Reviewer 2 Report

The review by Solimando et al. on Mast cells and interleukins is an interesting overview of the mast cells as a potential source of interleukins and their role in the regulation of biological activities. The authors describe the role of the IL-1, IL-2, IL-3, IL-4, IL-5, IL-6, IL-8, IL-9, IL-10, IL-11, IL-13, IL-16, IL-17 and IL-33 secreted by mast cell in the context of inflammatory disorders, autoimmune diseases and tumor growth. They suggested some therapeutic approaches. I have some specific comments that I would like to see addressed.

Major comments:

1.       Line 30: “Mast cells originate in the bone marrow microenvironment”. Recent works describe that mast cell can derived from erythro-myeloid progenitors which originate in the yolk sac before the emergence of hematopoietic stem cells. Please include this in the text.

2.       Line 95: “Mast cell- derived IL-3 promotes the differentiation and survival of other myeloid cells”. Please develop, which myeloid cells?

3.       Line 202-204: “IL-33 is a potent inducer of proinflammatory cytokine and chemokines, and it is produced by mast cells and regulates IgE-dependent inflammation [62]”. Please develop, give details about the cytokine produced.

4.       Line 231 “such as TNF-α and IL-4. Indeed, the IL-9 neutralization ” What is the relation between TNF-α, IL-4 and IL-9?

5.       Line 258: Table 1. The table 1 is poor. Please add mechanism or fact (eg. IL-8 secretion or increase mast cell number ect...). Add references as well.

6.        In table 1: Melanoma are skin cancer. Please choose or specify which type of skin cancer.

7.        Line 294-295: “Figure 2”. There is no figure 2. Please include it or delete it in the text.

8.       Some cytokines such as IL-31 and IL-36 are missing. Please add them in the text.

Minor points:

1.       Line 22: “Vertebrates” take out the majuscule

2.       Line 41: “related to blood and lymphatic vessels, and nerves, and are involved in”). Check grammar

3.       Line 50-51: “Mast cells can exert their multiple actions in many pathological conditions through a plethora of mediators contained in their cytoplasmic granules, which after mast cell stimulation are released in the extracellular space through different mechanisms of 51 degranulation”. Check grammar

4.       Line 54: “proteases, serotonin, heparin, proteases” take out one proteases

5.       Line 74-77: “IL-1 exerts a pro-inflammatory action, inducing an increase in the vascular permeability where it has been secreted, with a consequent increase in the concentration of leukocyte migration in the anatomical site where inflammation occurs.” Rephrase please.

6.       Line 81-82: Please move the phrase “Mast cell IL-2 contributes to the expansion of Treg cells involved in immune suppression in an experimental model of IL-33-induced airway inflammation [9]” at the end of the line 86.

7.       Line 102: “Mast cell-derived IL-4 activates of CD4+ T cells” check grammar.

8.       Line 106:”T cell and mast cell by meand” check spelling

9.       Line 150-158: different style, please homogenise.

10.   Line 202-204: “IL-33 is a potent inducer of proinflammatory cytokine and chemokines” add “s” at cytokine.

11.   Line 219: “TNF α I” put a”-“ before the “α”.

12.   Line 230: “pathogenesis of both the diseases.” Check grammar

13.   Line 244: “lymphoctes” check spelling, “y” is missing

14.   Line 247: “which allows for the development” check grammar.

15.   Line 327-342: The writing style is different, please homogenise

16.   Line 633: the reference 102 is not cited in the text

Author Response

REVIEWER 2

Major comments:

   Line 30: “Mast cells originate in the bone marrow microenvironment”. Recent works describe that mast cell can derived from myeloid progenitors which originate in the yolk sac before the emergence of hematopoietic stem cells. Please include this in the text.

      We have improved the text as follows: “Recent works describe that mast cell can derived from erythro-myeloid progenitors (EMPs) which originate in the yolk sac before the emergence of hematopoietic stem cells. In this context, perturbation of the maternal environment leads to the production of mediators such as cytokines that will lead to epigenetic alterations in  EMPs and thereby, chronic activation of EMP-derived immune cells, including mast cells, after birth [3]”. 

Line 95: “Mast cell- derived IL-3 promotes the differentiation and survival of other myeloid cells”. Please develop, which myeloid cells?

We have improved the text as follows: “Mast cell-derived IL-3 promotes the differentiation and survival of other myeloid cells, including basophils, eosinophils, and macrophages”.

Line 202-204: “IL-33 is a potent inducer of proinflammatory cytokine and chemokines, and it is produced by mast cells and regulates IgE-dependent inflammation [62]”. Please develop, give details about the cytokine produced.

We have specified that “IL-33 is a potent inducer of proinflammatory cytokine and chemokines ,

including IL-5,

IL-13, CXCL8 and CCL2, and it is produced by mast cells and regulates IgE-dependent inflammation [62]”.

Line 231 “such as TNF-α and IL-4. Indeed, the IL-9 neutralization ” What is the relation between TNF-α, IL-4 and IL-9?

We have specified that: “Mast cells produce cytokines that have been implicated in EAE disease, such as TNF-α and IL-4. In this context, T cells activate mast cells present within the leptomeninges and thedura mater, to produce and release TNF-α, which in turn stimulates other inflammatory cells to release IL-4 and IL-9.

Line 258: Table 1. The table 1 is poor. Please add mechanism or fact (eg. IL-8 secretion or increase mast cell number ect...). Add references as well.

We have improved the Table 1 according to Reviewer’s suggestion.

In table 1: Melanoma are skin cancer. Please choose or specify which type of skin cancer.

We have deleted the reference to skin cancer from the Table.

Line 294-295: “Figure 2”. There is no figure 2. Please include it or delete it in the text.

We have deleted the reference to figure 2.

Some cytokines such as IL-31 and IL-36 are missing. Please add them in the text.

We have improved the text as follows: IL-31 is a cytokine that is produced not only in T-cells but also in mast cells. It is implicated to play a key role in inflammatory diseases and in the pathogenesis of itch in atopic dermatitis. IL-31 induced CCL1, CCL17, and CCL22 chemokines in atopic dermatitis-irritated skin. IL-1, released by activated keratinocytes and mast cells, stimulates skin macrophages to release IL-36, a powerful pro-inflammatory IL-1 family member. IL-36 mediates both innate and adaptive immunity, including chronic pro-inflammatory diseases such as psoriasis. 

Reviewer 3 Report

The authors A.G.Solimando, V.Desantis and D.Ribatti in the publication "Mast cells and interleukins" present an interesting review that reveals the problem of interleukins in the aspect of mast cell biology. The use of interleukins by mast cells for the regulation of a specific tissue microenvironment is extremely important in the study of the fundamental mechanisms of adaptive responses and pathological conditions. However, some aspects of the review should be improved and supplemented with relevant information to form a more comprehensive portrait of the biology of interleukins in mast cells. The reviewer has several questions and suggestions.

1.      The authors need to supplement the data of their review with data from a number of publications on the biology of interleukins in mast cells:

·         Eissmann MF, Buchert M, Ernst M. IL33 and Mast Cells-The Key Regulators of Immune Responses in Gastrointestinal Cancers? Front Immunol. 2020 Jul 3;11:1389. doi: 10.3389/fimmu.2020.01389. PMID: 32719677; PMCID: PMC7350537.

·         Haque TT, Frischmeyer-Guerrerio PA. The Role of TGFβ and Other Cytokines in Regulating Mast Cell Functions in Allergic Inflammation. Int J Mol Sci. 2022 Sep 17;23(18):10864. doi: 10.3390/ijms231810864. PMID: 36142776; PMCID: PMC9503477.

·         Komi DEA, Mortaz E, Amani S, Tiotiu A, Folkerts G, Adcock IM. The Role of Mast Cells in IgE-Independent Lung Diseases. Clin Rev Allergy Immunol. 2020 Jun;58(3):377-387. doi: 10.1007/s12016-020-08779-5. PMID: 32086776; PMCID: PMC7244458.

·         Nagata K, Nishiyama C. IL-10 in Mast Cell-Mediated Immune Responses: Anti-Inflammatory and Proinflammatory Roles. Int J Mol Sci. 2021 May 7;22(9):4972. doi: 10.3390/ijms22094972. PMID: 34067047; PMCID: PMC8124430.

·         Robuffo I, Toniato E, Tettamanti L, Mastrangelo F, Ronconi G, Frydas I, Caraffa Al, Kritas SK, Conti P. Mast cell in innate immunity mediated by proinflammatory and antiinflammatory IL-1 family members. J Biol Regul Homeost Agents. 2017 Oct-Dec;31(4):837-842. PMID: 29254286.

2.      It is advisable for the authors to supplement their review with data on IL-24, information, which can be found in the work "Mitamura et al., 2020":

·         Mitamura Y, Nunomura S, Furue M, Izuhara K. IL-24: A new player in the pathogenesis of pro-inflammatory and allergic skin diseases. Allergol Int. 2020 Jul;69(3):405-411. doi: 10.1016/j.alit.2019.12.003. Epub 2020 Jan 21. PMID: 31980374.

3.      Строка 292. "Mast cells through the release of IL-1, IL-4 and IL-6 are involved in the elimination  of tumor cells and rejection of tumors [85]. IL-1 is linked to tumorigenesis and tumor  progression [86] and angiogenesis, being also a potential therapeutic target [87,88] (Figure  2).»

Authors should explain the conflicting data regarding the biological effect of IL-1 on tumors.

4.      Line 260: The authors discuss the role of mast cells in the anti-tumor effect of mast cells through direct contact: “In this context, an anti-tumor role of mast cells reflects the ability of these cells to mediate direct tumor killing. Otherwise, increased mast cell number has been correlated with a poor prognosis (???, Reviewer) in several human tumors.”

Here, the authors should discuss the results of a recent study on the mechanisms of antitumor activity of mast cells in melanoma, showing the interaction of mast cells with tumor cells in situ:

          Atiakshin D, Kostin A, Buchwalow I, Samoilova V, Tiemann M. Protease Profile of Tumor-Associated Mast Cells in Melanoma. Int J Mol Sci. 2022 Aug 11;23(16):8930. doi: 10.3390/ijms23168930. PMID: 36012196; PMCID: PMC9408654.

5.       Строка 277. Авторы пишут: «Finally, tryptase is responsible for both tumor cell disruption and inflammation by means of the activation of protease-activated receptors (PAR-1 and -2).» The authors should more in detail discuss the involvement of tryptase in inflammation. They can use a comprehensive overview on this issue:

·         Atiakshin D, Buchwalow I, Samoilova V, Tiemann M. Tryptase as a polyfunctional component of mast cells. Histochem Cell Biol. 2018 May;149(5):461-477. doi: 10.1007/s00418-018-1659-8. Epub 2018 Mar 12. PMID: 29532158.

6.      Line 269. The authors need to confirm with a reference the phrase: “However, pro-angiogenic factors exceeds the record of antiangiogenic factors”.

7.      Line 280. The authors write: “Two mast cell phenotypes have been described called MC1 and MC2, related to pro-inflammatory and anti-inflammatory profiles, respectively. The pro-inflammatory profile is characterized by low expression of IL-10, while the anti-inflammatory profile is characterized by high expression of IL-10 [82].” Here, the authors should pay attention to the existing classification of human mast cells according to the expression of specific proteases. 

8.              Can the authors provide information on the mechanisms of secretion of interleukins by mast cells?

9.              Can the authors clarify the information on how selective the process of interleukin secretion can depend on the state of a specific tissue microenvironment? The authors would like to consider the question of what other components of the mast cell secretome are associated with the secretion of interleukins.

10.            It is desirable to complement a review with the question of autoregulation of TC activity with the help of interleukins.

Author Response

REVIEWER 3

The authors need to supplement the data of their review with data from a number of publications on the biology of interleukins in mast cells:

 Eissmann MF, Buchert M, Ernst M. IL33 and Mast Cells-The Key Regulators of Immune Responses in Gastrointestinal Cancers? Front Immunol. 2020 Jul 3;11:1389. Doi: 10.3389/fimmu.2020.01389. PMID: 32719677; PMCID: PMC7350537.

Haque TT, Frischmeyer-Guerrerio PA. The Role of TGFβ and Other Cytokines in Regulating Mast Cell Functions in Allergic Inflammation. Int J Mol Sci. 2022 Sep 17;23(18):10864. Doi: 10.3390/ijms231810864. PMID: 36142776; PMCID: PMC9503477.

 Komi DEA, Mortaz E, Amani S, Tiotiu A, Folkerts G, Adcock IM. The Role of Mast Cells in IgE-Independent Lung Diseases. Clin Rev Allergy Immunol. 2020 Jun;58(3):377-387. Doi: 10.1007/s12016-020-08779-5. PMID: 32086776; PMCID: PMC7244458.

Nagata K, Nishiyama C. IL-10 in Mast Cell-Mediated Immune Responses: Anti-Inflammatory and Proinflammatory Roles. Int J Mol Sci. 2021 May 7;22(9):4972. Doi: 10.3390/ijms22094972. PMID: 34067047; PMCID: PMC8124430.

Robuffo I, Toniato E, Tettamanti L, Mastrangelo F, Ronconi G, Frydas I, Caraffa Al, Kritas SK, Conti P. Mast cell in innate immunity mediated by proinflammatory and anti-inflammatory IL-1 family members. J Biol Regul Homeost Agents. 2017 Oct-Dec;31(4):837-842. PMID: 29254286.

We have added these publications in the text and in the reference list.

It is advisable for the authors to supplement their review with data on IL-24, information, which can be found in the work “Mitamura et al., 2020”:  Mitamura Y, Nunomura S, Furue M, Izuhara K. IL-24: A new player in the pathogenesis of pro-inflammatory and allergic skin diseases. Allergol Int. 2020 Jul;69(3):405-411. Doi: 10.1016/j.alit.2019.12.003. Epub 2020 Jan 21. PMID: 31980374.

We have improved the text as follows: “IL-24 is a member of the IL-20 family, released by mast cells. It is involved in allergic airway inflammation, chronic allergic urticaria, allergic contact and atopic dermatitis.”

Line 292. “Mast cells through the release of IL-1, IL-4 and IL-6 are involved in the elimination of tumor cells and rejection of tumors [85]. IL-1 is linked to tumorigenesis and tumor progression [86] and angiogenesis, being also a potential therapeutic target [87,88]”. Authors should explain the conflicting data regarding the biological effect of IL-1 on tumors.

To avoid a contradictory statement, we have modified the sentence as follows: “IL-1 is linked to tumorigenesis and tumor progression in terms of local growth, angiogenesis, macrophage recruitment and metastatic process [86].

Line 260: The authors discuss the role of mast cells in the anti-tumor effect of mast cells through direct contact: “In this context, an anti-tumor role of mast cells reflects the ability of these cells to mediate direct tumor killing. Otherwise, increased mast cell number has been correlated with a poor prognosis (???, Reviewer) in several human tumors.”

Here, the authors should discuss the results of a recent study on the mechanisms of antitumor activity of mast cells in melanoma, showing the interaction of mast cells with tumor cells in situ:Atiakshin D, Kostin A, Buchwalow I, Samoilova V, Tiemann M. Protease Profile of Tumor-Associated Mast Cells in Melanoma. Int J Mol Sci. 2022 Aug 11;23(16):8930. doi: 10.3390/ijms23168930. PMID: 36012196; PMCID: PMC9408654.

We have modified the text as follows: “According, increased mast cell number has been correlated with a good or prognosis poor prognosis in several human tumors. Recently, Atiakshin and co-workers carried out a a detailed assessment of the mechanisms of biogenesis and excretion of mast cell proteases in melanoma. They demonstrated that the development of melanoma was accompanied by the appearance in the tumor-associated mast cell population of several pools with a predominant content of one or two specific proteases with a low content or complete absence of others.In this context, a sufficient number of mast cells expressing specific proteases in the tumor stroma apparently belong to a protective mechanism against tumor evasion from immunocompetent cells and can serve as a favorable prognosis.”

Finally, tryptase is responsible for both tumor cell disruption and inflammation by means of the activation of protease-activated receptors (PAR-1 and -2).» The authors should more in detail discuss the involvement of tryptase in inflammation. They can use a comprehensive overview on this issue: Atiakshin D, Buchwalow I, Samoilova V, Tiemann M. Tryptase as a polyfunctional component of mast cells. Histochem Cell Biol. 2018 May;149(5):461-477. doi: 10.1007/s00418-018-1659-8. Epub 2018 Mar 12. PMID: 29532158.

We have improved the text as follows: “Tryptase is involved in the pathogenesis of different inflammatory diseases of the cardiovascular, respiratory, digestive, reproductive and nervous systems, and its participation in the development of these pathological conditions is the consequence of the crivation of PAR-2 receptors by proteases”.

Line 269. The authors need to confirm with a reference the phrase: “However, pro-angiogenic factors exceeds the record of antiangiogenic factors”.

We have added a reference.

 Line 280. The authors write: “Two mast cell phenotypes have been described called MC1 and MC2, related to pro-inflammatory and anti-inflammatory profiles, respectively. The pro-inflammatory profile is characterized by low expression of IL-10, while the anti-inflammatory profile is characterized by high expression of IL-10 [82].” Here, the authors should pay attention to the existing classification of human mast cells according to the expression of specific proteases. 

To avoid overlapping with the existing classification of human mast cells according to the expression of specific proteases, we have modified the sentence and I have deleted the mention of the two mast cell phenotypes called MC1 and MC2 as follows: “Mast cells may be polarized through a pro-inflammatory and anti-inflammatory profiles, respectively. Pro-inflammatory profile is characterized by low expression of IL-10, while anti-inflammatory profile is characterized by high expression of IL-10 [82]”.

Can the authors provide information on the mechanisms of secretion of interleukins by mast cells?

We have added information concerning the mechanisms of secretion of IL-4 and IL-15: “ It is unclear whether mast cell IL-4 released extracellularly is due to degranulation or constitutive exocytosis. Microvesicle sheeding is one of the secretory pathways for the release of IL-15 in mast cells

Can the authors clarify the information on how selective the process of interleukin secretion can depend on the state of a specific tissue microenvironment? The authors would like to consider the question of what other components of the mast cell secretome are associated with the secretion of interleukins.

We have improved the texts as follows: ILs are key regulators of tumor microenvironment and are responsible of tumor–immune cell crosstalk. In this context, ILs are relevant in the development and progression of cancer. Although various immune effector cells are recruited to the tumor site, including mast cells, their antitumor functions are downregulated, largely in response to tumor-derived factors or signals.The multitude of cellular sources, including mast cells, receptors and signaling pathways are responsible of pleiotropic role of ILs in cancer. Different components of the mast cell secretome, including secreted de novo synthesized soluble molecules, secreted preformed granules, and the membrane-enclosed extracellular vesicles, are involved in the release of ILs in the tumor microenvironment.

It is desirable to complement a review with the question of autoregulation of TC activity with the help of interleukins.

We apologize, but we don’t’ understand the significance of TC acronyms.

Round 2

Reviewer 2 Report

The authors have satisfactorily addressed all my concerns

Author Response

I thanks The Reviewer for his/her positive evaluation.